# Trapping and Methanation of CO$_2$ in a Domestic Microwave Oven Using Combinations of Sorbents and Catalysts

Loren Acher *, Tristan Laredo, Thierry Caillot, Akim Kaddouri and Frederic C. Meunier *

Univ Lyon, Université Claude Bernard Lyon 1, CNRS, IRCELYON, 2 Av. Albert Einstein,
69626 Villeurbanne, France; tlaredocontact@gmail.com (T.L.); thierry.caillot@ircelyon.univ-lyon1.fr (T.C.);
akim.kaddouri@ircelyon.univ-lyon1.fr (A.K.)
* Correspondence: loren.acher@univ-lyon1.fr (L.A.); fcm@ircelyon.univ-lyon1.fr (F.C.M.)

**Abstract:** CO$_2$ trapping and methanation allow to reduce greenhouse gas emissions and recycle CO$_2$ into a sustainable fuel, provided renewable H$_2$ is employed. Microwave (MW)-based reactors provide an efficient means to use electrical energy for upgrading chemicals, since MW can selectively heat up the load placed in the reactor and not the reactor itself. In this study, CO$_2$ capture and methanation were investigated using solid adsorbents (ZrO$_2$ and Fe$_3$O$_4$), microwave absorbers (SiC and Fe$_3$O$_4$) and Ru/SiO$_2$ as CO$_2$ the methanation catalyst. The sorption and catalyst beds were located in a domestic MW oven that was used to trigger CO$_2$ desorption and methanation in the presence of H$_2$. The working Fe-based structure turned out to be a mixture of FeO and Fe, which allowed for MW absorption and local heating; it also acted as a CO$_2$ sorbent and reverse water–gas shift catalyst. Various reactor configurations were used, leading to different performances and selectivity to CO and CH$_4$. To the best of our knowledge, this is the first report of its kind showing the potential of using inexpensive microwave technology to readily convert trapped CO$_2$ into valuable products.

**Keywords:** carbon capture and utilization; synthetic natural gas; microwave-assisted desorption; methane





## 1. Introduction

CO$_2$ emissions represent a major fraction of greenhouse gas emissions induced by human activities and should be reduced to limit climate change [1]. CO$_2$ capture and utilization (CCU) will play a prominent role in restructuring the energy sector and the fight against global warming. CO$_2$ could be trapped within large industrial units (e.g., power plants, boilers, and cement manufacture) and reprocessed on-site or remotely. The hydrogenation of CO$_2$ into convenient fuels such as CH$_4$ through the Power-to-Gas (PtG) technology is of strong interest [2]. CH$_4$ is produced through Sabatier reaction (Equation (1)), which is catalyzed by Ru- or Ni-based catalysts [3].

$$CO_2 + 4\,H_2 \rightarrow CH_4 + 2\,H_2O \tag{1}$$

$$CO_2 + H_2 \rightarrow CO + H_2O \tag{2}$$

Note that the reverse water–gas shift (RWGS) reaction (Equation (2)) is often proposed as an intermediate step during methanation [4]. The H$_2$ needed for these reactions should preferably be obtained from water electrolysis using renewable or nuclear energy. CH$_4$ could be then directly injected into the already existent pipeline networks or storage infrastructures. Developing an effective system that could combine CO$_2$ capture and conversion to CH$_4$ in a single unit is of particular interest for the efficiency of PtG plants.

Amine solutions are used industrially for CO$_2$ trapping [5], and a recent process was proposed that could produce CH$_4$ at 170 °C in a multiphasic batch reactor by mixing CO$_2$-saturated amine solutions with a Ru-based heterogeneous catalyst [6]. Yet, processing high-pressure multiphasic batch reactors is costly and complex. Simpler isothermal sequential

processes solely based on solid traps and catalysts, so-called "dual function materials" (DFMs), have been proposed [7–10]. DFMs contain a $CO_2$-storing component, typically an alkali or Earth-alkali oxide, which exhibit a strong basic character [11–13], and a catalytic metal able to hydrogenate $CO_2$ to methane, usually Ru or Ni or alloys of those [7–10]. Na-exchanged zeolites exhibit some of the highest $CO_2$ adsorption capacity under direct air capture conditions [14].

The release of the trapped $CO_2$ implies an energy cost, which can be minimized by using oxides of weaker basicity such as zirconia [15]. Energy savings can also be achieved by using microwave (MW) heating, which does not require heating up the whole reactor assembly [16]. Ellison et al. recently reported that the use of microwave heating for $CO_2$ desorption from zeolite 13X increased the adsorption/desorption cycling productivity and potentially reduced the energy penalty of sorbent regeneration [17]. In addition, these authors determined an apparent activation energy of the MW-assisted desorption which is significantly lower than that obtained via conventional heating. Jang et al. also showed that desorption of a solid guanidine carbonate under MW heating is up to 17 times faster than conventional conductive heating at 160 °C, resulting in a 40% electrical energy reduction [18]. Moreover, energy savings for desorption can rise up to 94% when the sorbent is microwave-transparent (i.e., when only the adsorbed molecules are heated up) [19]. Tubes, fitting and reactors can be made of microwave-transparent materials such as polytetrafluoroethylene PTFE (=Teflon®) or quartz [20,21], meaning that those also would not consume electromagnetic energy and remain at low temperatures. Such a design should allow for further energy saving and facilitate process design.

We recently reported on the use of a simple domestic MW oven to carry the combined trapping and combustion of volatile organic compounds (i.e., toluene, n-decane and formaldehyde) [22]. The method and experimental setup used in this previous study is actually well-suited to carry out similar work on the trapping and methanation of $CO_2$. This study will actually be the first of its kind, to the best of our knowledge, as we could not find any previous investigation on the combined trapping and methanation of $CO_2$ in an MW oven. Earlier MW-based studies involving $CO_2$ and $CH_4$ were mostly focused on methane dry reforming [20].

The objective of the present study was to investigate the possibility of using a domestic microwave oven to desorb $CO_2$ from a sorbent under MW irradiation and simultaneously hydrogenate the released $CO_2$ to methane. High surface area $ZrO_2$ was used as $CO_2$ adsorbent because of its known basicity [15] and high thermal stability. The use of Na-exchanged zeolite was avoided in the present study because of the potential temperature runaway under MW that can lead to melting of these solids [22]. SiC and $Fe_3O_4$ were employed as MW absorbers [20]. Silica-supported ruthenium was chosen as methanation catalyst because of its high activity [8–10] and the ease with which it can be reduced. The results will show that the various extents of $CO_2$ conversion and selectivity to $CH_4$ can be obtained depending on bed configurations, requiring to further optimize the system to maximize the yield to methane and limit bed overheating that led to catalyst deactivation.

## 2. Materials and Methods

$ZrO_2$ (monoclinic, 131 $m^2$ $g^{-1}$) was provided by MEL Chemicals and characterized in detail elsewhere [23]. $Fe_3O_4$ (10 $m^2$ $g^{-1}$) was bought from Alfa Aesar (lot X08G023). SiC (120 grit, ca. 137 μm particles) was purchased from Alfa Aesar (batch #10179265). A silica-supported ruthenium methanation catalyst (6.8 wt.% Ru measured with ICP, Ru dispersion = 22% measured using $H_2$ chemisorption, corresponding to 4 nm particles) was prepared via dry impregnation using $RuCl_3$-x$H_2O$ precursor. High purity gases $N_2$, $H_2$, $O_2$ and $CO_2$ from Air Liquide were used without any further purification.

The experimental setup is illustrated in Figure 1A,B. The gas flow rates were controlled using Brooks electronic mass flow controllers. A commercial 800 W MW domestic oven (Sharp, model YC-MS01E-W) was modified as described elsewhere [22]. The MW oven power was always set to the maximum when used. Two quartz tubes were placed in the

oven to accommodate the sorbent and catalyst beds, using PTFE (Teflon®, Wilmington, NC, USA) Swagelok fittings. Sorbent and catalyst beds were held with quart wool plugs. Gases were flown through polyethylene tubing passing through 2 mm holes drilled in the oven cavity.

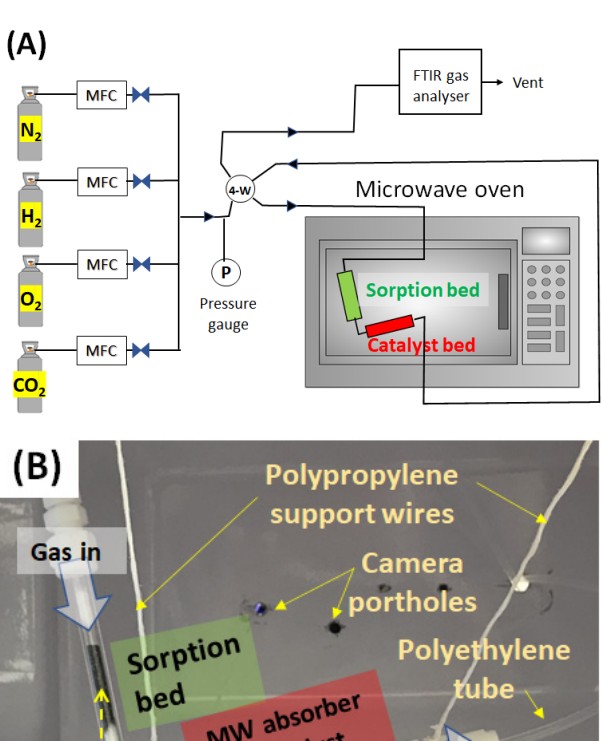

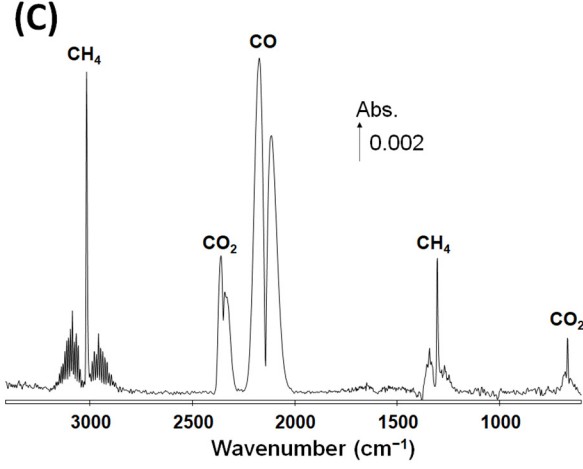

**Figure 1.** (**A**) Schematic representation of the catalytic setup used for $CO_2$ trapping and methanation. Gases were fed through mass flow controllers (MFCs). The feed could be sent through a 4-way valve (4-W) to the sorbent and catalytic beds located in the MW oven first or directly into the FT-IR gas analyzer. (**B**) Picture of the sorption and catalyst beds located in the MW oven. (**C**) Examples of FT-IR transmission spectra showing the typical signals of methane, CO and $CO_2$ measured with the IR gas cell.

The feed could be sent through the sorbent and catalytic beds first or directly into the IR gas analyzer, which consisted of a 10 cm pathlength IR gas cell (from Harrick) fitted in a Bruker Tensor 27. The gas cell was kept at 80 °C and 8 scans were averaged at a resolution of 4 cm$^{-1}$. A typical FT-IR transmission spectrum showing methane, CO and $CO_2$ bands are given in Figure 1C. IR Band assignment was ascertained via comparison to the NIST database [24–26]. $CO_2$ and product concentrations were derived from calibration curves, using the area of the bands located at 670 cm$^{-1}$ (for $CO_2$), 3015 cm$^{-1}$ (for methane) and 2145 cm$^{-1}$ (for CO). The conversion of $CO_2$ was calculated as the ratio between reacted $CO_2$ and the inlet $CO_2$, the reacted $CO_2$ being the difference between the measured $CO_2$ and the inlet $CO_2$. The signal of the inlet $CO_2$ was measured by by-passing the reactor. The yield of CO (or $CH_4$) was calculated as the concentration of CO (or $CH_4$) divided by that of reacted $CO_2$.

Zones in which an effective MW absorption by the materials occurred were determined from sample black body radiation, since it is difficult to measure temperatures in multimode MW ovens and because of the likely presence of strong temperature gradients within long (several centimeters) beds [27,28]. Two positions associated with intense MW fields were found, shown in Figure 1B, expressed by the distance (in mm) from the oven center and height: (50; 30) and (115; 60), respectively. Figure 2 shows typical brilliances obtained from a mixture of SiC and $Fe_3O_4$ placed in the catalyst bed. It is worth stressing that the SiC + $Fe_3O_4$ mixture became red hot (i.e., T > 500 °C) in less than 10 s.

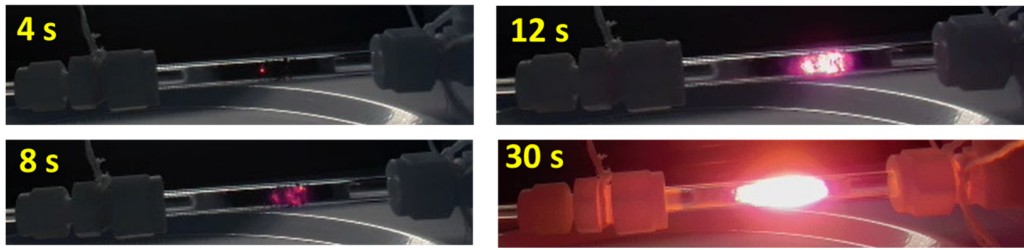

**Figure 2.** Pictures of a physical mixture of 354 mg of SiC and 91 mg of $Fe_3O_4$ under a $N_2$ stream after various times under 800 W MW irradiation. A minute-long video of the sequence is available at the following link: https://mycore.core-cloud.net/index.php/s/rhQAXNZGD5O7qq7 (accessed on 2 November 2023).

An *operando* diffuse reflectance FT-IR spectroscopy (DRIFTS) cell (High Temperature reaction cell from Harrick, described elsewhere [29]) was used to measure the activity for $CO_2$ methanation of the fresh and aged Ru catalysts as a function of temperature and determine the nature of surface species present under reaction conditions. The catalyst temperature was measured with a thermocouple located at the top of the bed within the thin catalyst layer (corresponding to precisely weighted masses around 10 mg) placed on SiC, since DRIFTS reaction cells often present large temperature gradients [29,30].

### 3. Results and Discussion

#### 3.1. $CO_2$ Methanation under Continuous Flow Conditions under MW

The sorption bed was left empty in this case to measure the activity of various combinations of SiC, $Fe_3O_4$ and $Ru/SiO_2$ loaded in the catalytic reactor under a continuous flow of $CO_2$ and $H_2$ while irradiated by MW. The presence of only SiC led to no measurable activity (Figure 3A). A full conversion of $CO_2$ was obtained over the SiC + $Fe_3O_4$ physical mixture, but the only product was CO. This observation is consistent with the well-known fact that iron oxides are active catalysts for the RWGS reaction (Equation (2)) but not for $CO_x$ methanation [31].

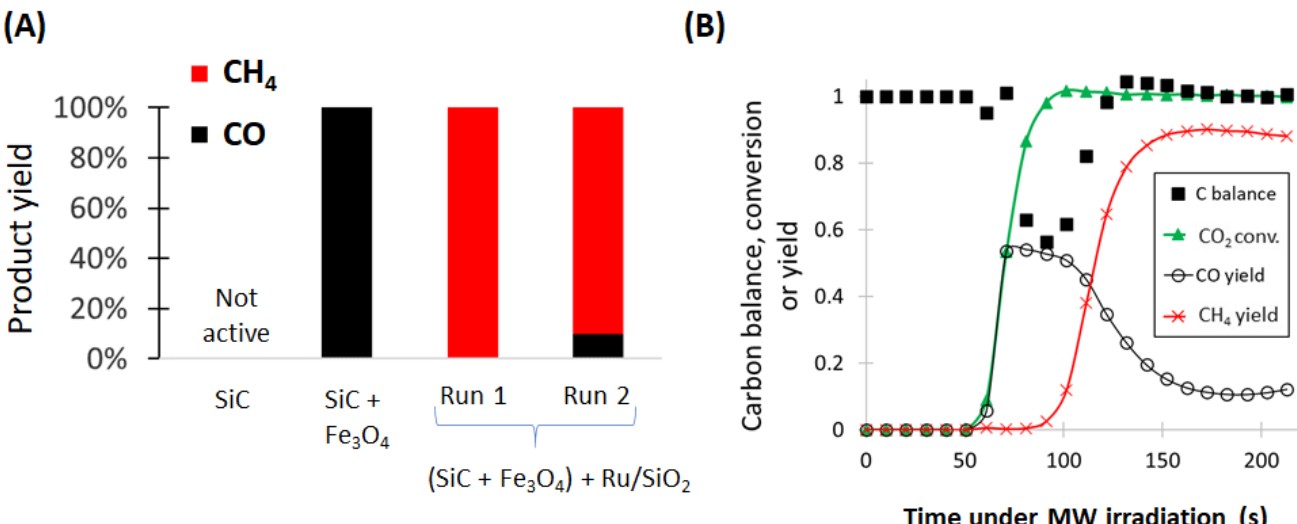

**Figure 3.** (**A**) Product selectivity at a steady-state under MW irradiation during continuous flow experiments. Catalyst masses used for the various runs (1) 450 mg SiC, (2) mixture of 354 mg SiC + 91 mg $Fe_3O_4$, (3) mixture of 354 mg SiC + 91 mg $Fe_3O_4$ followed by 25 mg $Ru/SiO_2$ and (4) same as (3). (**B**) Carbon balance, $CO_2$ conversion and product yields as a function of time under MW irradiation during the second run with SiC + $Fe_3O_4$ + $Ru/SiO_2$. Feed: 2% $CO_2$ + 20% $H_2$ in $N_2$, total flow: 50 mL $min^{-1}$, MW power = 800 W.

A layer of $Ru/SiO_2$ was added to catalytic reactor just after the SiC + $Fe_3O_4$ mixture, separated with a 5 mm quartz wool plug. This configuration led to a full $CO_2$ conversion, with methane being the only product at steady-state (Figure 3A, third entry). However, the second run with the same configuration led to a $CH_4$ yield of about 90% and a yield of CO of 10%. This suggests that the Ru-based catalyst had been deactivated during the first run, likely due to sintering of the Ru particles exposed to the high temperatures of the red-hot glowing SiC + $Fe_3O_4$ mixture, despite being located about 5 mm away.

The time-resolved product analysis of the second run with the (SiC + $Fe_3O_4$) + $Ru/SiO_2$ configuration is shown in Figure 3B. The initial lag time before observing any conversion was both due to the time needed to purge the line dead-volume to reach the IR analyzer and the time needed to heat up the reactor under MW. CO was initially observed as the sole reaction product, before declining in yield, followed by the rise of the methane yield. This suggests that the initial activity was only due to the RWGS activity of the $FeO_x$ component and that the temperature of the Ru catalyst was too low to achieve methanation.

Interestingly, the carbon balance was significantly lower than 100% just after the light off (Figure 3B). This suggests that part of the CO formed was initially trapped in the reactor, likely as carbonyl species formed at the surface of the Ru nanoparticles. This hypothesis was checked by carrying out an *operando* DRIFTS analysis described in the subsequent section.

### 3.2. $CO_2$ Methanation under Continuous Flow Conditions in a DRIFTS Reactor

The *operando* study was carried out by introducing the methanation stream (2% $CO_2$ + 20% $H_2$ in $N_2$) at room temperature over the samples without any pre-treatment to mimic typical CCU operations under which the sample is initially exposed to $CO_2$ and air. The Ru nanoparticles were thus likely oxidized. No formation of carbonyl of Ru could be observed at room temperature (Figure 4A), indicating that the Ru surface was oxidized or could not activate $CO_2$.

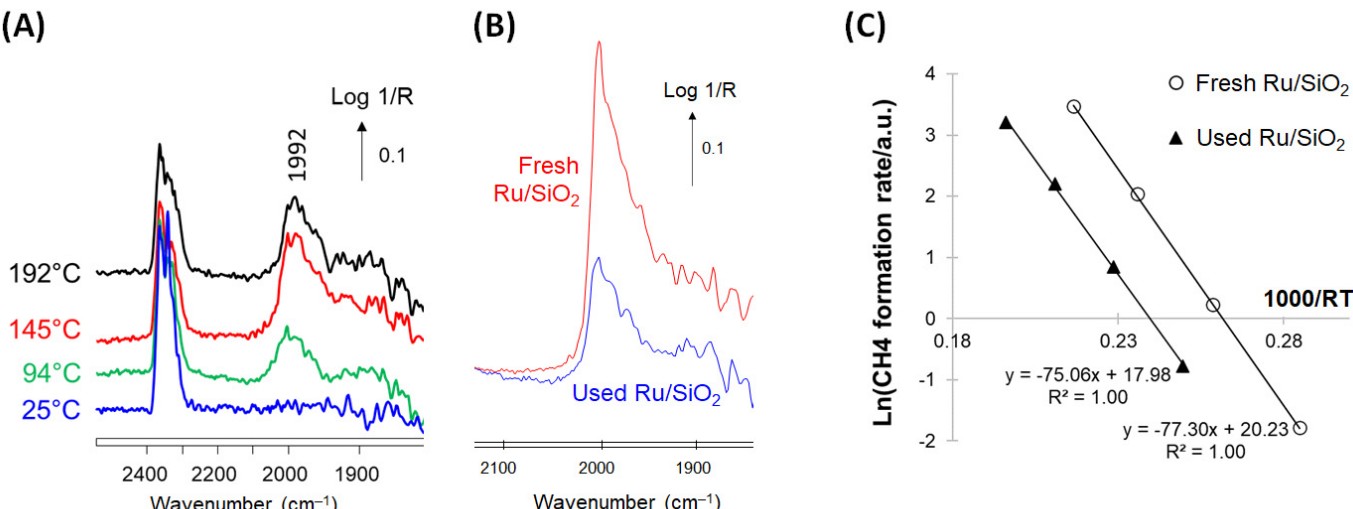

**Figure 4.** (**A**) *Operando* DRIFTS spectra collected over the used $Ru/SiO_2$ sample under 2% $CO_2$ + 20% $H_2$ in $N_2$ at various temperatures. A KBr spectrum was used as background. (**B**) Comparison of the *operando* DRIFTS spectra of the fresh and used $Ru/SiO_2$ samples under 2% $CO_2$ + 20% $H_2$ in $N_2$ at 180 °C. The spectra were normalized to silica overtones to ensure identical optical pathlengths. (**C**) Corresponding Arrhenius-like plots of the natural logarithm of the rate of methane formation against the reciprocal temperature for both the fresh and used $Ru/SiO_2$ catalysts.

No methanation activity could be measured below 100 °C neither over the fresh nor on the used sample, although bands assigned to $Ru^0$-CO carbonyls [32] were clearly observed at 94 °C (Figure 4A). This indicates that at 94 °C Ru was partly reduced and able to dissociate $CO_2$ and store CO. This likely explains the uncomplete carbon balance observed around 90 °C during the flow experiment (Figure 3B), as the carbonyls formed poisoned the surface at low temperatures.

The ca. 4-fold lower DRIFTS band area of carbonyls species observed over the used sample as compared to the fresh catalyst (Figure 4B) and ca. 5-fold lower methanation rate indicated by the shifted Arrhenius plots (Figure 4C) suggest that the deactivation was mostly due to sintering of the Ru nanoparticles, likely induced by the high temperature of the neighboring radiating SiC + $Fe_3O_4$ when exposed to MW. Sintering is a common process in heterogeneous catalysis and can occur through various mechanisms, e.g., metal particle coalescence and Ostwald ripening [33], depending on the mobility of atoms and thermodynamic stability of the particles with respect to their size. Different strategies can be employed to limit sintering, e.g., using porous supports strongly interacting with the metal particles. The sintering of materials exposed to MW has also been discussed in detail elsewhere [34]

Note that the apparent activation energy remained unchanged at about 76 kJ mol$^{-1}$, in agreement with those reported for highly loaded Ru-based catalysts [35].

### 3.3. $CO_2$ Trapping Followed by Desorption and Methanation under MW

In the following pulsed experiments, $CO_2$ adsorption was carried out at room temperature using a 2% $CO_2$ + 20% $O_2$ in $N_2$ for 5 min, which was always sufficient to saturate the sorbent used. The system was then purged under $N_2$ for 5 min, then left for 5 min under 20% $H_2$ in $N_2$ with a total flow of 50 mL min$^{-1}$. The MW power was then turned on for 5 min still under 20% $H_2$ in $N_2$ with a total flow of 50 mL min$^{-1}$, while monitoring the gases evolved from the system.

$ZrO_2$ poorly absorbed MW and did not release $CO_2$ if not mixed with SiC. SiC did not adsorb and release any $CO_2$ when used alone. In the first pulse experiment shown, the sorption bed was filled with a mixture of $ZrO_2$ + SiC, while the catalyst bed contained both a mixture of SiC + $Fe_3O_4$ as MW absorbers, followed by $Ru/SiO_2$ acting as a methanation

catalyst (Figure 5A). Figure 5B shows the compounds desorbed as a function of time during MW irradiation. Only a small concentration of $CO_2$ was released, followed by CO and then a larger peak of methane.

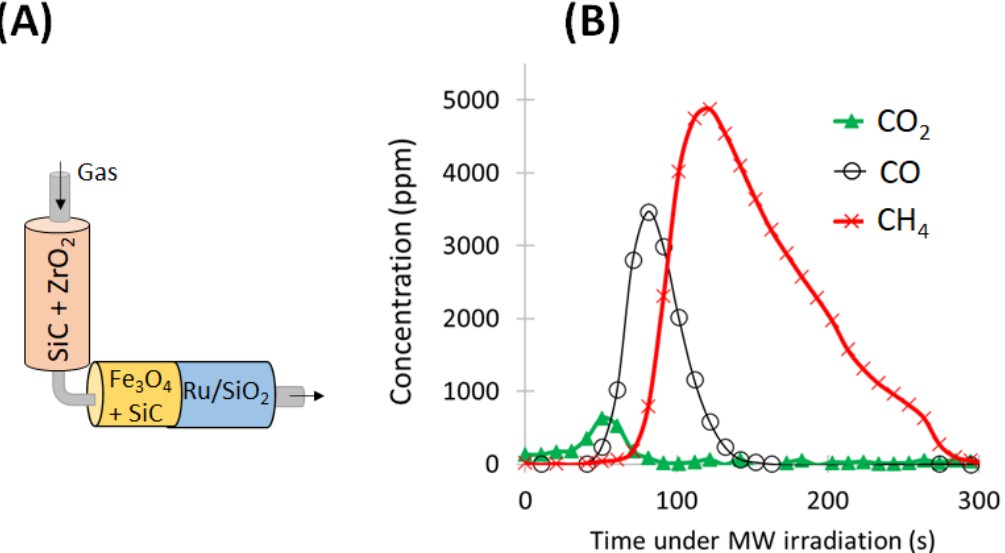

**Figure 5.** (**A**) Configuration used: 350 mg of SiC + 32 mg of $ZrO_2$ were placed in the sorption bed and a mixture made of 354 mg of SiC + 91 mg of $Fe_3O_4$ followed by 25 mg of $Ru/SiO_2$ were placed in the catalyst bed. (**B**) Products released as a function of time under MW. The sample was first exposed to 2% $CO_2$ + 20% $O_2$ in $N_2$ for 5 min, then purged under $N_2$ for 5 min, then left for 5 min under 20% $H_2$ in $N_2$ with a total flow of 50 mL min$^{-1}$. The MW were then turned on for 5 min still under 20% $H_2$ in $N_2$ with a total flow of 50 mL min$^{-1}$.

These data are consistent with the steady flow experiment reported in Figure 3, in which CO appeared as a primary reaction product, followed by methane. The heating rate of the Ru catalyst seemed insufficient to convert the CO formed initially via RWGS (Equation (2)) over the $Fe_3O_4$.

Experiments were carried out to determine the amount of equivalent C (i.e., sum of CO, $CO_2$ and $CH_4$) released by the main components of the system described in Figure 5A. As mentioned earlier, SiC did not release any carbon-containing compound (Figure 6) and the same was observed with the $Ru/SiO_2$ catalyst. Interestingly, $Fe_3O_4$ led to a large amount of products, about 3-fold higher than the $ZrO_2$. It must be concluded that the Fe constituent formed some carbonate species when exposed to the $CO_2$ + $O_2$ stream.

The $CO_2$ released by zirconia under our conditions was about 150 μmol g$^{-1}$, while the use of amine-based methods can lead to reversible sorption capacity as high as 3500 μmol g$^{-1}$ [36]. It is therefore clear that the adsorption on solids is unlikely to match that obtained in amine solutions. Some of the interests of solids lie in easier handling, lower toxicity and higher thermal stability.

The gain in terms of released species obtained using a sorbent bed with $ZrO_2$ appeared thus limited. A limited interest was also observed in terms of product selectivity by comparing a series of experiment with and without the sorption bed (Figure 7). Similar high $CO_2$ conversion and selectivity to methane were obtained in both cases.

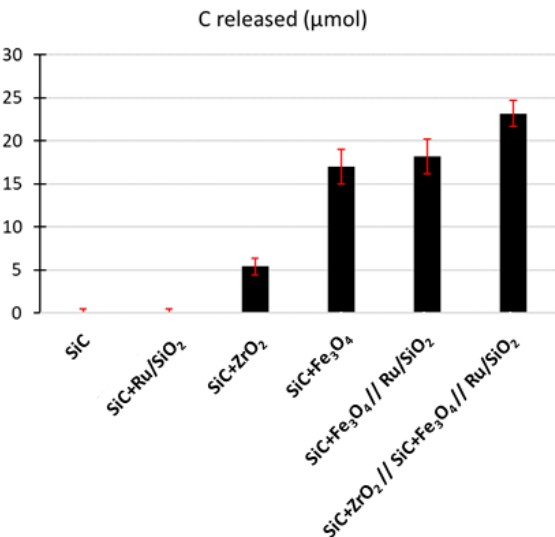

**Figure 6.** Amount of carbon released (i.e., sum of CO, $CO_2$ and $CH_4$) by the various elements of the sorbent–catalyst system described in Figure 5A.

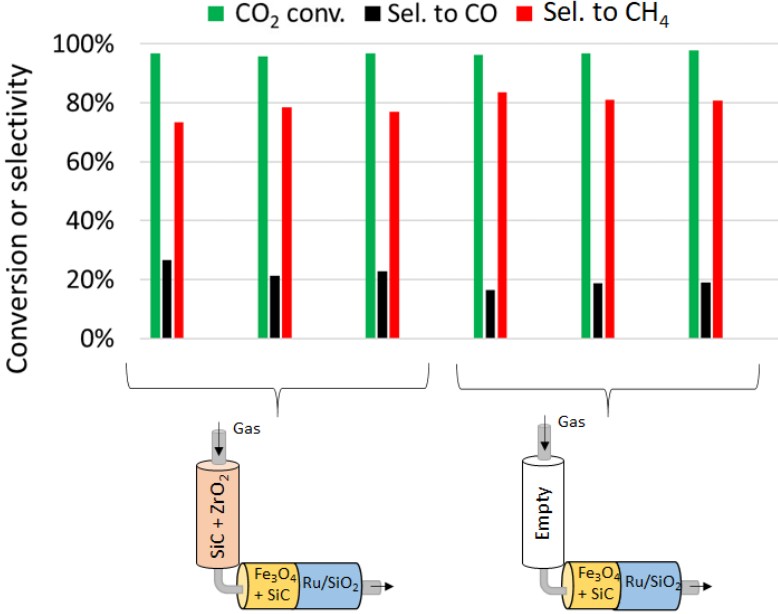

**Figure 7.** $CO_2$ conversion and selectivity to CO and methane obtained over a configuration (**Left**) including and (**Right**) lacking a sorption bed. The reaction sequence and catalyst bed are the same as those detailed in Figure 5.

The significant proportion of CO (ca. 20%) in the reaction products is not desirable, and a different catalyst bed configuration was attempted by mixing directly the Ru catalyst with the iron oxide (which can heat up under MW in the absence of SiC) and zirconia to favor fast reduction and higher activity of Ru (Figure 8A). SiC was left out to limit excess temperature, and the mixture thus used became hot under MW irradiation but never glowed red.

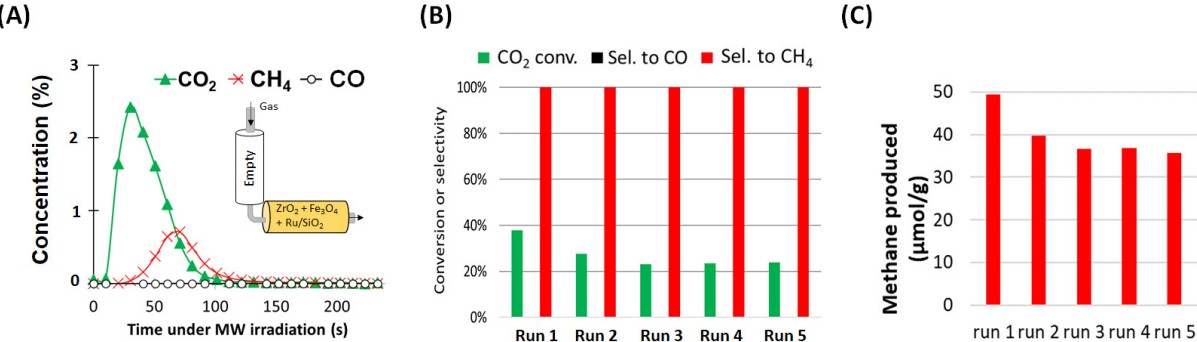

**Figure 8.** Experiment using a catalyst bed made of a mixture of 111 mg of $Fe_3O_4$ + 146 mg of $ZrO_2$ and 28 mg of $Ru/SiO_2$. The sorbent bed was empty. (**A**) Products released as a function of time under MW. (**B**) $CO_2$ conversion and selectivity to CO and methane over five consecutive runs. (**C**) Corresponding production of methane per total mass of the bed. The reaction sequence is the same as that detailed in Figure 5.

The desorption of compounds was fast and essentially completed in 100 s (Figure 8A), but unreacted $CO_2$ was the main species released. Nonetheless, no CO was observed over five consecutive runs, $CH_4$ being the only product (Figure 8B). The quantity of methane produced per total bed mass somewhat declined before stabilizing to about 35 µmol g$^{-1}$ over the five runs. This value is about 10-fold lower than those reported by Farrauto over some of the best dual function materials (DFM) operated in conventional ovens [8]. Yet, we believe that this is promising for a system that has not yet been optimized and offers a lot of parameters to adjust (e.g., sorbent, MW-absorber, catalyst, bed configuration, and MW power).

The nature of the iron phases present after use were determined with XRD analyses (Figure 9). The used Fe-based material, collected after the experiment reported in Figure 7, contained metallic iron [37,38] and FeO [39] as the main phases, with only traces of $Fe_3O_4$ [40] left. These two phases (metallic iron and FeO) are known to absorb better MW than Fe$^{III}$ compounds, as discussed elsewhere [41–44]. It is thus important to avoid full oxidation of the iron phases. The abilities of the FeO phase to store $CO_2$ as a carbonate and to absorb MW are much greater than that of metallic iron, thus stabilizing FeO appears important for future investigations.

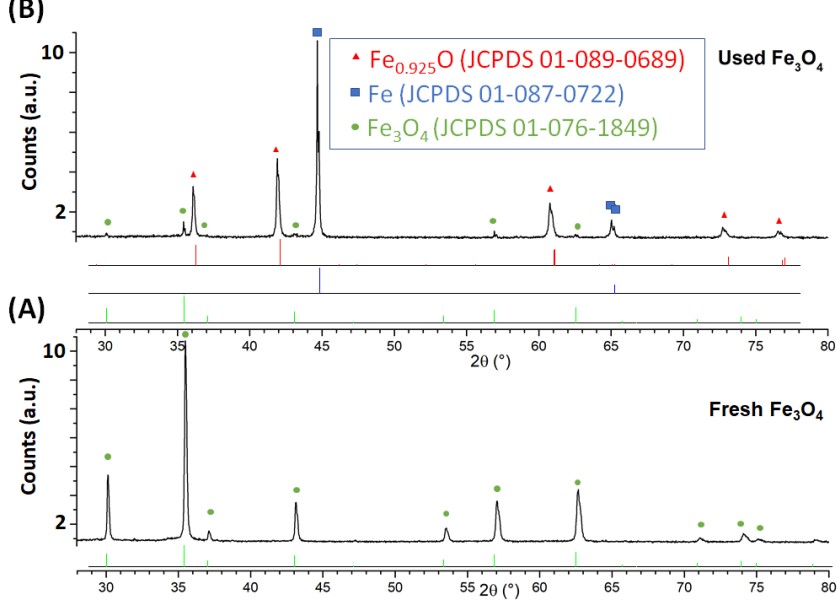

**Figure 9.** XRD pattern of the (**A**) fresh and (**B**) used $Fe_3O_4$ material.

The present study reports for the first time, to the best of our knowledge, that $CO_2$ desorption and methanation can be readily achieved in a domestic microwave oven. The economics of the process would need to be estimated in detail, but probably once the sorbent, MW absorber, methanation catalyst and the process conditions would have been optimized further from the present proof of concept. Since an earlier study reported up to 94% energy savings for an MW-based desorption processes [19], the present method can be regarded as highly promising that could lead to an inexpensive trapping–methanation technology.

## 4. Conclusions

$CO_2$ trapping and methanation were investigated over sorbents ($ZrO_2$ and $Fe_3O_4$) and MW absorbers (SiC and $Fe_3O_4$), using $Ru/SiO_2$ as a $CO_2$ methanation catalyst. $Fe_3O_4$ adsorbed significant concentration of $CO_2$ but overheated up above 600 °C under MW, leading to a partial sintering of neighboring Ru catalyst. The working Fe-based structure turned out to be a mixture of FeO and Fe, while the $Fe_3O_4$ phase essentially disappeared. The reactor configuration with a sorption bed filled with $SiC+ZrO_2$ followed by a catalytic bed filled with SiC + $Fe_3O_4$ ahead of a $Ru/SiO_2$ layer led to almost full $CO_2$ conversion with a significant production of CO (ca. 20%) along with $CH_4$. Using only a catalytic reactor filled with a physical mixture of $ZrO_2$ + $Fe_3O_4$ + $Ru/SiO_2$ allowed for fast desorption (<2 min) and full selectivity to $CH_4$, although $CO_2$ conversion was only about 20%.

In conclusion, it is possible to use an inexpensive reactor system to trap/release and methanate $CO_2$ under MW irradiation. The proportions and configurations of sorbent(s), MW absorber(s) and methanation catalysts can be adapted to control product selectivity, $CO_2$ conversion, catalyst deactivation and process timescale.

**Author Contributions:** Conceptualization: All authors. Methodology: F.C.M.; validation: all authors. All authors have read and agreed to the published version of the manuscript.

**Funding:** This research received no external funding.

**Institutional Review Board Statement:** Not applicable.

**Data Availability Statement:** The data presented in this study are available on request from the corresponding author. The data are not publicly available due to file size constraints.

**Acknowledgments:** The IRCELYON Scientific Services are acknowledged for carrying out the XRD measurements.

**Conflicts of Interest:** The authors declare no conflict of interest.

## Glossary

| | |
|---|---|
| DFM | Dual function materials |
| DRIFTS | Diffuse reflectance Fourier-transform infra red |
| MW | microwave |
| PTFE | polytetrafluoroethylene = Teflon® |
| PtG | Power-to-Gas technology |
| RWGS | reverse water-gas shift reaction |

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
