# Peer review of "Trapping and Methanation of CO2 in a Domestic Microwave Oven Using Combinations of Sorbents and Catalysts"

_applsci, doi:10.3390/app132312536_

Round 1

Reviewer 1 Report

Comments and Suggestions for Authors

This is a very interesting work that focuses on CO2 capture and methanation technology. The data of the experiment are very detailed, and the logic of the manuscript is also recognized. The article has certain novelty and innovation and has practical significance. However, there are still some shortcomings that require more in-depth discussion before publication.

1. The abstract and keywords can be further examined and refined to highlight the core of this work.

2. How to check the yield of CO2 and methane, please describe in detail.

3. In the description of FT-IR, all characteristic peaks should be referenced. Please refer to the FT-IR description of this article (https://doi.org/10.1016/j.cclet.2023.108813).

4. In Fig.3, Whether the author has calculated the carbon balance.

5. In Fig.6, the error bar is too large, and the experimental error is not allowed to exceed 3% under normal circumstances.

6. Please supplement the reference for the characteristic peaks in XRD.

7. The quality of the current picture in this paper is difficult to meet the requirements of journal publication, please aesthetic

Comments on the Quality of English Language

English need to be further polished.

Author Response

Thank you for the global positive view of our manuscript.

1- We have slightly modified the abstract and the keywords. 

2- We have added the definition of the product yields, which reads: “CO2 and product concentrations were derived from calibration curves, using the area of the bands located at 670 cm-1 (for CO2), 3015 cm1 (for methane) and 2145 cm1 (for CO). The conversion of CO2 was calculated as the ratio between reacted CO2 and the CO2 fed, the reacted CO2 being the difference between the measured CO2 and the CO2 fed. The CO2 fed was measured by-passing the reactor. The yield of CO (or CH4) was calculated as the concentration of CO (or CH4) divided by that of reacted CO2.”

3- Unfortunately, our institutions do not have access to the paper suggested by the reviewer. We have therefore simply cited the NIST database that is freely accessible and added the following sentence: “IR Band assignment was ascertained by comparison to the NIST database. [CO: https://webbook.nist.gov/cgi/cbook.cgi?ID=C630080&Units=SI&Type=IR-SPEC&Index=1#IR-SPEC  CH4: https://webbook.nist.gov/cgi/cbook.cgi?ID=C74828&Units=SI&Type=IR-SPEC&Index=1#IR-SPEC CO2 : https://webbook.nist.gov/cgi/cbook.cgi?ID=C124389&Units=SI&Type=IR-SPEC&Index=1#IR-SPEC ]”

4- We have added the carbon balance to Figure 3, which makes the discussion clearer and added the following sentence “Interestingly, the carbon balance was significantly lower than 100 % just after the light off (Figure 3B).”

5- The error bars in Figure were estimated from the standard deviation obtained upon repeated measurements. It turns out indeed to be higher than 3 % in the present case, and was intrinsically related to errors in the MW absorption efficiency and data analysis.

6- the details of the original references related to the JCPDF files of the XRD patterns have been added.

[] Hull, A. W.  Crystal Structure of Iron. Phys. Rev., 1917, 10, 661.

[] Shull, C. G.; Wilkinson, M. K. Neutron Diffraction Studies of Various Transition Elements. Rev. Mod. Phys. 1953, 25(1), 100–107. https://doi.org/10.1103/RevModPhys.25.100

[] Fjellvåg, H.; Grønvold, F.; Stølen, S.; Hauback, B. On the Crystallographic and Magnetic Structures of Nearly Stoichiometric Iron Monoxide. J. Solid State Chem. 1996 , 124(1), 52–57. https://doi.org/10.1006/jssc.1996.0206

[] Dvoryankina, G. G.;  Pinsker, Z. G.  Electron diffraction pattern investigation of Fe3O4Dokl. Akad. Nauk SSSR1960, 132(1), 110–113.

7- The pictures have been improved, but we would be happy to further improve those if specific points are raised.

Reviewer 2 Report

Comments and Suggestions for Authors

Overall, the manuscript is very well written and the author(s) presented their experimental work in a proper way. The main comment is on the scalability of this research. The presented experimental work is used for a very small quantity of CO2 but a large amount is required in real sense. What will be the application? The author (s) have not elaborated on this, which is the main limitation of this research. If, authors highlight this limitation or discuss the scalability then it will be more beneficial for the reader.

Kindly add a flow chart for the methodology to make it easy for the reader to understand.

(Lines – 9 – 10): “: CO2 trapping and methanation provides……assuming sustainable H2 is employed.” Here “capturing” is a suitable word rather than trapping because “trapping” is usually used for injecting CO2 in the subsurface. Kindly rephrase this sentence.

(Line – 12) “……..heat up loads of interest and not the reactor” This is an ambiguous statement and in the abstract, it is hard to understand what the researcher wants to say. Kindly rephrase it in such a way that reader can easily understand. If mention the name of load, it will be easier to understand.

 (Line – 79) In Figure – 1, (A), a 4-way valve is used. Why do gases after the MW go back to the 4-way valve rather than directly go to the FTIR? Is this 4-way valve being NR valve?

Author Response

Thank you for these comments. The present paper is a proof-of-concept and stresses that desorption and methanation can be carried fast in a MW oven. Applicability will indeed depend on the quantities of CO2 to treat and this will be detailed in a subsequent paper.

We have improved and enlarged the figures so that the reader can better understand the methodologies used. We feel that the flow charts already present in figure 1, 5 and 7 are now sufficient to guide the reading.

We have rephrased the sentences raised by the reviewer, thank you for pointing out these.

Using a 4-way valve allows to go directly to the gas analyser or through the reactors. By-pass measurement would not be possible if the MW vent was directly connected to the gas analyser.

Reviewer 3 Report

Comments and Suggestions for Authors

The manuscript entitled "Trapping and methanation of CO2 in a domestic microwave oven using combinations of sorbents and catalysts" developed a new procedure for the trapping and methanation of CO2 over sorbents (ZrO2, Fe3O4) and MW absorbers (SiC, Fe3O4), using Ru/SiO2 as CO2 methanation catalyst. After careful study, I recommend the revision of manuscript following to "Major Revision". The comments are presented as follows:

1)     The main objective of this manuscript must be explained in last paragraph of introduction.

2)     The chemical structures and reaction mechanism of primary, secondary and tertiary amines with CO2 must be explained in detail.

3)      Is trapping and methanation of CO2 over sorbents and MW absorbers cost-effective? A techno-economic analysis must be added.

4)     English of the manuscript needs revision.

5)     Use the following article and cite it to enrich the introduction. https://doi.org/10.1371/journal.pone.0236529

6)     A nomenclature list should be added to the manuscript to cover all the acronyms.

7)     Introduce the most efficient sorbents/absorbers for CO2 methanation and trapping.

Comments on the Quality of English Language

moderate edition is needed

Author Response

Thank you for the constructive comments.

1- We have modified the introduction to better stress the objective of the present study by adding the following sentence in the last paragraph: “The objective of the present study was to investigate the possibility of using a domestic microwave oven to desorb CO2 from a sorbent under MW irradiation and simultaneously hydrogenate the released CO2 to methane.”.

2- The structure of the amine used to trap CO2 are certainly very important, but less relevant to the present study that uses inorganic solid traps. We would therefore prefer not to discuss amine sorbents.

3- This is a very important point raised by the reviewer indeed and at the moment we do not have a precise answer that would require a full analysis that is outside the scope of the present study. An earlier study already mentioned in the introduction suggested energy saving up to 94% for desorption when the sorbent is MW-transparent. We have added the following sentence just before the conclusion section: “The present study reports, for the first time to the best of our knowledge, that CO2 desorption and methanation can be readily achieved in a domestic microwave oven. The economics of the process would need to be estimated in details, but probably once the sorbent, MW-absorber, methanation catalyst and the process conditions would have been optimized further from the present proof-of-concept. Since an earlier study reported up to 94 % energy savings for a MW-based desorption processes [18], the present method can be regarded as highly promising that could lead to an inexpensive trapping-methanation technology.”

4- The English has been improved.

5- We would be happy to add relevant citations, but there must be an error because the proposed link refers to a paper entitled “Modification of polyethersulfone membrane using MWCNT-NH2 nanoparticles and its application in the separation of azeotropic solutions by means of pervaporation”, which have no connection to our manuscript.

6- A glossary has been added at the end of the manuscript.

7- We have added two sentences in the introduction section and a recent review discussing this matter, which reads “Na-exchanged zeolites exhibit some of the highest CO2 adsorption capacity under direct air capture conditions [Lai, J.Y.; Ngu, L.H.; Hashim, S.S. A review of CO2 adsorbents performance for different carbon capture technology processes conditions. Greenhouse Gas Sci. Technol. 2021, 11, 1076-1117. https://doi.org/10.1002/ghg.2112].” and “ The use of Na-exchanged zeolite was avoided in the present study because of the potential temperature runaway under MW that can lead to melting these solids [22].”

Reviewer 4 Report

Comments and Suggestions for Authors

The article "Trapping and methanation of CO2 in a domestic microwave oven using combinations of sorbents and catalysts" by Acher L., et. al. explores a novel approach to capture and convert carbon dioxide (CO2) into methane (CH4) using a domestic microwave oven, solid sorbents, and catalysts. The study investigates a novel approach to CO2 capture and conversion, which has the potential to reduce greenhouse gas emissions and recycle CO2 effectively. It focuses on using a domestic microwave oven as a reactor for CO2 trapping, desorption, and methanation, employing various solid sorbents (ZrO2, Fe3O4) and catalysts (Ru/SiO2), as well as microwave-absorbing materials (SiC, Fe3O4). The researchers experiment with different reactor configurations to assess their performance, CO2 conversion, and selectivity to CO and CH4. The article is well written, thoroughly supported by data and should be accepted in this journal if the authors can address following questions-

1.     The article mentions that microwave heating can reduce energy consumption compared to conventional methods. Can you provide more quantitative data or analysis on the energy savings achieved with this approach, especially in comparison to other CO2 conversion technologies?

2.     How does the performance of ZrO2, used as a CO2 sorbent in the study, compare with traditional amine-based solutions commonly used for CO2 capture in terms of efficiency and energy consumption?

3.     Can you elaborate on the mechanisms and factors causing catalyst sintering, and what strategies could be employed to mitigate this issue while maintaining high catalytic activity?

4.     The article discusses the challenge of catalyst deactivation due to high temperatures. Can you explain in detail how the sintering of Ru particles occurs and its impact on the CO2 methanation process?

5.     Fe3O4 is mentioned as a material that absorbs CO2 but overheats under microwave irradiation. Can you delve into the underlying chemical and physical processes that lead to this behavior, and how might it be mitigated in future research?

Author Response

Thank you for these very positive and constructive comments.

1- This very relevant question was also raised by the second reviewer and we reproduce the corresponding answer: “This is a very important point raised by the reviewer indeed and at the moment we do not have a precise answer that would require a full analysis that is outside the scope of the present study. An earlier study already mentioned in the introduction suggested energy saving up to 94% for desorption when the sorbent is MW-transparent. We have added the following sentence just before the conclusion section: “The present study reports, for the first time to the best of our knowledge, that CO2 desorption and methanation can be readily achieved in a domestic microwave oven. The economics of the process would need to be estimated in details, but probably once the sorbent, MW-absorber, methanation catalyst and the process conditions would have been optimized further from the present proof-of-concept. Since an earlier study reported up to 94 % energy savings for a MW-based desorption processes [18], the present method can be regarded as highly promising that could lead to an inexpensive trapping-methanation technology.”

2- This is indeed an important point and we have added the following sentences to the discussion: “The CO2 released by zirconia under our conditions was about 150 µmol g1, while the use of amine-based methods can lead to reversible sorption capacity as high as 3500 µmol g–1 [Liu, F.; Jing, G.; Zhou, X.; Lv, B.; Zhou, Z. Performance and Mechanisms of Triethylene Tetramine (TETA) and 2-Amino-2-methyl-1-propanol (AMP) in Aqueous and Nonaqueous Solutions for CO2 ACS Sustainable Chemistry & Engineering 2018; 6, 1352-1361. https://pubs.acs.org/doi/10.1021/acssuschemeng.7b03717]. It is therefore clear that the adsorption on solids is unlikely to match that obtained in amine solutions. Some of the interests of solids lie in easier handling, lower toxicity and thermal stability.

The energy consumption issue was dealt with the answer to reviewer 2, reproduced here: “This is a very important point raised by the reviewer indeed and at the moment we do not have a precise answer that would require a full analysis that is outside the scope of the present study. An earlier study already mentioned in the introduction suggested energy saving up to 94% for desorption when the sorbent is MW-transparent. We have added the following sentence just before the conclusion section: “The present study reports, for the first time to the best of our knowledge, that CO2 desorption and methanation can be readily achieved in a domestic microwave oven. The economics of the process would need to be estimated in details, but probably once the sorbent, MW-absorber, methanation catalyst and the process conditions would have been optimized further from the present proof-of-concept. Since an earlier study reported up to 94 % energy savings for a MW-based desorption processes [18], the present method can be regarded as highly promising that could lead to an inexpensive trapping-methanation technology.”

3- Sintering is unfortunately a common process in heterogeneous catalysis and can occur through various mechanisms, e.g. metal particle coalescence and Ostwald ripening, depending on the mobility of atoms and thermodynamic stability of the particles with size. Different strategies can be employed to limit sintering, e.g. porous support strongly interacting with the metal particles. We have added the following sentences and references in the discussion section: “Sintering is a common process in heterogeneous catalysis and can occur through various mechanisms, e.g. metal particle coalescence and Ostwald ripening [Hansen, T.W.; DeLaRiva, A.T.; Challa, S.R.; Datye, A.K. Sintering of Catalytic Nanoparticles: Particle Migration or Ostwald Ripening? Accounts of Chemical Research 2013, 46, 1720-1730], depending on the mobility of atoms and thermodynamic stability of the particles with respect to their size. Different strategies can be employed to limit sintering, e.g. porous support strongly interacting with the metal particles. The sintering of materials exposed to MW has also been discussed in details elsewhere [Rybakov, K.I. ; Olevsky, E.A. ; Krikun, E.V. Microwave Sintering: Fundamentals and Modeling. J. Am. Ceram. Soc., 2013, 96, 1003-1020. https://doi.org/10.1111/jace.12278]”.

4- In the present case we do not have any details about the mode Ru sintered and this would be important to further investigate. The lower methanation activity is then due to the lower concentration of surface Ru atoms.

5- This is a crucial point and our data shows that the heating form of the sample corresponded to a mixture of FeO and Fe. We have added the following sentence and references that discuss in detail the heating power of MW on materials depending on their structure: “These two phases (metallic iron and FeO) are known to absorb better MW than Fe3+ compounds, as discussed elsewhere [Microwave-Enhanced Chemistry. Fundamentals, Sample Preparation and Applications Edited by H. M. Kingston (Duquesne University) and Stephen J. Haswell (University of Hull). American Chemical Society:  Washington, D.C. 1997. xxviii + 772 pp. $109.95. ISBN 0-8412-3375-6 ; Buchelnikov, V.D. ; Louzguine-Luzgin, D.V. ; Xie, G. ; Li, S. ; Yoshikawa, N. ; Sato, M. ; Anzulevich, A.P. ; Bychkov, I.V. ; Inoue, A. Heating of metallic powders by microwaves: Experiment and theory. Appl. Phys. 2008,  104, 113505. https://doi.org/10.1063/1.3009677 ; Tanaka, M. ; Kono, H. ; Maruyama, K. Selective heating mechanism of magnetic metal oxides by a microwave magnetic field. Phys. Rev. B 2009, 79, 104420 ; McGill, S.L. ; Walkiewicz, J.W. ; Smyres, G.A. The Effects of Power Level on the Microwave Heating of Selected Chemicals and Minerals. MRS Online Proceedings Library 1988, 124, 247–252. https://doi.org/10.1557/PROC-124-247]. It is thus important to avoid full oxidation of the iron phases.”

Round 2

Reviewer 3 Report

Comments and Suggestions for Authors

acceptable

Comments on the Quality of English Language

minor edition